# Dynamics of X chromosome hyper-expression and inactivation in male tissues during stick insect development

Jelisaveta Djordjevic[1]*, Patrick Tran Van[1,2], William Toubiana[1], Marjorie Labédan[1], Zoé Dumas[1], Jean-Marc Aury[3], Corinne Cruaud[4], Benjamin Istace[3], Karine Labadie[4], Benjamin Noel[3], Darren J. Parker[1,5], Tanja Schwander[1]*

1 Department of Ecology and Evolution, University of Lausanne, Lausanne, Switzerland, 2 Institut Curie, PSL Research University, INSERM, Paris, France, 3 Génomique Métabolique, Genoscope, Institut François Jacob, CEA, CNRS, Univ Evry, Université Paris-Saclay, Evry, France, 4 Genoscope, Institut François Jacob, CEA, Université Paris-Saclay, Evry, France, 5 School of Environmental and Natural Sciences, Bangor University, Bangor, United Kingdom

* jelisaveta.djordjevic@unil.ch (JD); tanja.schwander@unil.ch (TS)

## Abstract

Differentiated sex chromosomes are frequently associated with major transcriptional changes: the evolution of dosage compensation (DC) to equalize gene expression between the sexes and the establishment of meiotic sex chromosome inactivation (MSCI). Our study investigates the mechanisms and developmental dynamics of dosage compensation and meiotic sex chromosome inactivation in the stick insect species *T. poppense*. Stick insects are characterized by XX/X0 sex determination, with an X chromosome that likely evolved prior to the diversification of insects over 450 Mya. We generated a chromosome-level genome assembly and analyzed gene expression from various tissues (brain, gut, antennae, leg, and reproductive tract) across developmental stages in both sexes. Our results show that complete dosage compensation is maintained in male somatic tissues throughout development, mediated by upregulation of the single X chromosome. Contrarily, in male reproductive tissues, dosage compensation is present only in the early nymphal stages. As males reach the 4th nymphal stage and adulthood, X-linked gene expression diminishes, coinciding with the onset of meiosis and MSCI, which involves classical silencing histone modifications. These findings reveal the dynamic regulation of X-linked gene expression in *T. poppense*, and suggest that reduced X-expression in insect testes is generally driven by MSCI rather than an absence of dosage compensation mechanisms. Our work provides critical insights into sex chromosome evolution and the complex interplay of dosage compensation and MSCI across tissues and developmental stages.

## Author summary

Males and females have often different numbers of sex chromosomes. For example, in humans, females have two X chromosomes, while males have one X and one Y. To avoid

---

**Data availability statement:** Genome assembly: PRJNA1126215; annotation and scripts to analyze data (https://github.com/JelisavetaDjordjevic/Dosage_compensation); raw sequence reads have been deposited in NCBI's sequence read archive under the following bioprojects: PRJEB76439 (reference genome), PRJNA1128554 (RNAseq reads), PRJNA1213497 (ChIP-seq), PRJNA1212995 (Hi-C). Data were processed to generate plots and statistics using R v3.4.4.

**Funding:** This study was funded by the European Research Council Consolidator Grant (ERC) (No Sex No Conflict to T.S.), the Swiss FNS grant 31003A_182495 (T.S.), Genoscope, the Commissariat à l'Énergie Atomique et aux Énergies Alternatives (CEA), France Génomique (ANR-10-INBS-09–08). The funders had no role in study design, data collection and analysis, decision to publish, or preparation of the manuscript.

**Competing interests:** The authors declare that they have no competing interests.

potential imbalances in gene expression from the sex chromosomes between males and females, many species have evolved dosage compensation mechanisms. However, little is known about how dosage compensation varies across tissues and during development. We investigated this variation in the stick insect *Timema poppense*, which has an X0 system where males have one X chromosome and females have two (there is no Y chromosome). Our analysis of gene expression across tissues and developmental stages revealed consistent dosage compensation in somatic tissues but variation in testes. While dosage compensation is complete in early nymphal testes, it is replaced by X chromosome inactivation in later developmental stages and adulthood, resulting in imbalanced expression levels between sexes. These findings provide new insights into how sex chromosome regulation evolves across tissues and developmental stages.

## Introduction

In species where sex is determined by differentiated sex chromosomes, sex chromosome copy number differs between males and females. For example, in X0 and XY systems, males have only one copy of each gene located on the X chromosome, while females have two. Because gene copy number generally correlates with expression levels [1], the expression level of X-linked genes in males is expected to be half that of the same genes in females. However, selection can act on the regulation of expression in a sex-specific manner, and potentially restore similar expression levels in both sexes [2–4], a process known as dosage compensation [2,5,6]. The mechanisms underlying dosage compensation between the sexes vary with respect to whether sex chromosomes are downregulated in the homogametic sex, or upregulated in the heterogametic sex. Downregulation of X-linked expression in females underlies dosage compensation in eutherian mammals and the worm *Caenorhabditis elegans*: In eutherian mammals, females inactivate one of the X chromosomes in each cell [6,7], whereas in *C. elegans*, the expression of both X chromosome copies in hermaphrodites is reduced to achieve similar total expression levels as for the single X in males [8,9]. Upregulation of the single X in males appears to underlie dosage compensation in diverse insect species [10–13]. The ancestral sex determination system in insects was most likely X0/XX or XY/XX male heterogamety [14]. There is considerable conservation of X chromosome gene content in many insect orders, suggesting that there may be an ancient insect X chromosome which partially persisted for over 450 MYA [15]. However, X upregulation in males is documented for both, species with a conserved X chromosome content as well as species with novel X chromosomes [11,13,16–18], where it involves different molecular mechanisms for different species [19,20].

Interestingly, in all studied X0 and XY species that feature dosage compensation via X upregulation in males, the single X in males is much less expressed in the testes than in the somatic tissues [11–13,21]. This is thought to be due to the presence of dosage compensation in somatic tissues and the absence of dosage compensation in the reproductive tissues, perhaps because the gene expression optima in male versus female gonads are so different that there is no or reduced selection for dosage compensation in this tissue [11]. Furthermore, transcriptional silencing of the X because of meiotic sex chromosome inactivation (MSCI) can specifically reduce the expression levels of X-linked genes in the testes of some species [22,23].

Beyond the specific case of testes in adult males, the extent of dosage compensation variation across tissues and development remains understudied [21,24]. Furthermore, the key mechanisms affecting X-linked expression in males (dosage compensation and meiotic sex chromosome inactivation) are only characterized for a few well-studied model organisms,

preventing inferences of conserved versus lineage-specific aspects of sex chromosome expression. Here we fill these knowledge gaps by exploring how the extent of dosage compensation varies over development in somatic and reproductive tissues as well as the potential impact of meiotic sex chromosome inactivation in the stick insect *Timema poppense*. These stick insects are characterized by XX:X0 sex determination [25] and an X chromosome content that is shared across many insect orders [15]. Previous work on *Timema* stick insects showed that in adults, there is dosage compensation in the head and legs, but not in the reproductive tract [13]. We generate a new chromosome-level genome assembly of *T. poppense* based on long-read sequencing and Hi-C scaffolding, and sequence RNA from brain, gut, antennae, leg and reproductive tract samples in males and females across development. We then assess the expression levels of genes located on the X chromosome(s) and autosomes in both sexes. Finally, we test for meiotic sex chromosome inactivation in the male germline via immunostaining and genome profiling for silencing histone marks, and discuss how MSCI globally affects X-linked gene expression patterns in male gonads.

## Materials and methods

### Sample collection and reference genome generation

We used wild-collected females of the species *T. poppense* (36°59'44.5"N 121°43'05.0"W) for generating the reference genome. We assembled contigs based on Nanopore and Illumina libraries (sequenced at 75x respectively 50x coverage) generated from a single female, and then scaffolded contigs using a Hi-C library (sequenced at 61x coverage) based on a different female. We annotated this assembly using transcriptome data from different tissues and development stages of male and female *T. poppense* as well as from a closely related species (*T. douglasi*; see Gene annotation section).

**Sequencing libraries.** To extract high molecular weight (HMW) DNA, we flash-froze a single female (without gut) in liquid nitrogen and ground it using a Cryomill (Retsch). We then extracted HMW DNA using a G/20 Genomic Tips kit (Qiagen) following the manufacturer's protocols. We checked DNA integrity on a pulse field agarose gel.

A total of four ONT libraries were prepared following Oxford Nanopore instructions. Four libraries were prepared using the SQK-LSK109 ligation sequencing kit and were loaded on four PromethION R9.4.1 Flow Cells. Flow Cell loading was performed according to the Oxford Nanopore protocol and resulted in 75X coverage.

A PCR free Illumina library (based on the same HMW extraction) was prepared using the Kapa Hyper Prep Kit (Roche, Basel, Switzerland), following manufacturer instructions. The library was quantified by qPCR using the KAPA Library Quantification Kit for Illumina Libraries (Roche), and the library profile was assessed using a High Sensitivity DNA kit on an Agilent Bioanalyzer (Agilent Technologies, Santa Clara, CA, USA). The library was then sequenced to approximately 50x coverage on an Illumina HiSeq 4000 instrument (Illumina, San Diego, CA, USA), using 150 base-length read chemistry in a paired-end mode.

A Hi-C library was prepared using the Proximo Hi-C Kit. We generated ground tissue for cross-linking using Cryomill (Retsch) and following manufacturer instructions, using a different female than the one used for Nanopore and HiSeq sequencing, from the same natural population. Library construction and sequencing (250 Mio read pairs) was outsourced to Phase Genomics (Seattle).

**Assembly pipeline and parameters.** Raw Oxford Nanopore reads were filtered using Filtlong v0.2.0 (https://github.com/rrwick/Filtlong) with the parameters --min_length 1000 --keep_percent 90 --target_bases 69050000000. The filtered Nanopore reads were then assembled into contigs using Flye v2.8.1 [26] with --genome-size 1.3 Gbp. All Nanopore

reads were mapped against the contigs using minimap2 v2.19 [27] with the parameters -c -x map-ont and a first step of polishing was performed using Racon v1.4.3 [28]. Three additional rounds of polishing were then conducted using the Illumina short reads. The short reads were aligned to the contigs using BWA mem v0.7.17 [29] and polishing was performed using Pilon v1.23 [30].

The assembly was decontaminated using BlobTools v1.0 [31] under the taxrule "bestsumorder". Hit files were generated after a blastn v2.10.1+ against the NBCI nt database, searching for hits with an e-value below 1e-25 (parameters: -max_target_seqs 10 -max_hsps 1 -evalue 1e-25). Contigs without hits to metazoans were removed. Haplotypic duplications were filtered out: filtered reads were mapped against the decontaminated genome using minimap2 and haplotigs were detected with Purge Haplotigs v1.1.1 [32] using the parameters -l 3 -m 17 -h 190 -j 101 following the recommendations by [32].

For scaffolding, Hi-C reads were mapped to the haploid genome using Juicer v1.6 [33] with the restriction site Sau3AI. Chromosome-level scaffolding was then performed using 3D-DNA v180922 [34] with the parameters --editor-coarse-resolution 25000, as recommended by the authors. The resulting Hi-C contact matrices were visualized with Juicebox, and polished following the recommendations by [33]. The completeness of the assembly was assessed with BUSCO v5.1.2 [35] and the insecta_odb10 dataset using the --long and --augustus parameters.

To identify the X chromosome in our assembly, we used a coverage approach. We compared coverage between males and females because *Timema* have XX/X0 sex determination [25] and males are expected to show half of the female coverage at the X chromosome. We mapped 3 female (SRS7637462, SRS7637490, and SRS7638308 from [36]) and 4 male samples (PRJNA725673 from [13]) to our scaffolded genome which allowed us to unambiguously identify the third largest scaffold as the X chromosome (S1 Fig).

**Gene annotation.** The *T. poppense* genome was annotated using a combination of *ab initio* gene prediction, protein homology, and RNA-seq using the Braker2 pipeline v. 2.1.6 [37]. To begin, the genome assembly was soft-masked using RepeatModeler (v. 2.0.2, options: -LTRStruct, -engine ncbi) and RepeatMasker (v. 4.1.2, options: -engine ncbi, -xsmall). For protein evidence, we used the arthropod protein sequences from OrthoDB v.10.1 [38] and the predicted protein sequences for *Timema* from our previous genome assemblies [36]. For RNAseq evidence, we used our newly generated RNAseq data for *T. poppense* (175 libraries, see below) as well as publicly available RNAseq data from *T. poppense* and the closely related species *T. douglasi* (from [39]) (Bioproject Accessions: PRJNA380865, PRJNA679785, PRJNA1128519) for a total of 376 RNAseq libraries (364 paired-end, 12 single-end) covering 117 different life stages, tissue, and sex combinations. Reads were quality trimmed with Trimmomatic (v. 0.39, options: ILLUMINACLIP:3:25:6 LEADING:9 TRAILING:9 SLIDINGWINDOW:4:15 MINLEN:80) [40] before mapping to the genome assembly with STAR (v. 2.7.8a, options: --twopassMode Basic) [41]. Braker2 was run using protein evidence and RNAseq separately with the gene predictors Augustus v. 3.4.0 [42] and Genemark v. 4.72 [43]. Following the RNAseq run, UTR predictions were added to the RNAseq gene predictions using GUSHR v. 1.0 [44] in Braker2 (--addUTR=on). The separate gene predictions were then merged using TSEBRA v1.0.3 [45] using the pref_braker1.cfg configuration file, which weights RNAseq evidence more strongly than the default option. We then ran BUSCO (v. 5.3.2, insecta_odb10) on the gene regions annotated by Braker2 and on the whole genome assembly. Any genes found by BUSCO but missed by Braker were then added to the annotation (48 genes). ncRNA genes were predicted using Infernal (v. 1.1.2, minimum e-value 1e-10) [46]. GO-terms for protein-coding gene predictions were obtained using blastP within OmicsBox v3.1.2, default parameters) to blast the nr *Drosophila melanogaster* database (taxonomy filter: 7227).

## Insect husbandry

Hatchlings were obtained from eggs laid by captive bred *T. poppense* individuals, originally collected in California in 2018. To complete development, *Timema* females have one extra molt compared to males [47], hence our aim was to obtain five different tissues (reproductive tracts and four somatic tissues) for each of seven developmental stages in females (nymphal stage 1-6, adult) and six in males (nymphal stage 1-5, adult; S1 Table). Upon hatching, insects were reared in Petri dishes containing *Ceanothus* plant cuttings wrapped in wet cotton up to a specific developmental stage and then dissected one day after molting. To identify individuals that molted, we painted the thoraxes with red acrylic paint after each molt and checked daily for individuals without the paint. Prior to dissection, the insects were anesthetized with $CO_2$. Brain, antennae, legs, and gut samples were obtained from each developmental stage (Fig 1A), while the reproductive tract was collected at every stage from the 4th nymphal stage (S1 Table). We dissected reproductive tracts only starting from the 4th nymphal stage because we were originally unable to unambiguously identify gonad tissues in the earlier stages. Upon improving our dissection techniques, we were later able to identify and dissect gonad tissues at earlier stages, and we therefore secondarily added reproductive tract samples of newly hatched individuals (1st nymphal stage) (Fig 1A). During dissections, tissues were placed in Eppendorf tubes and immediately flash frozen in liquid nitrogen before storage at -80°C until extraction for approximately 1/3 of the dissections. For the remaining dissections, due to pandemic-related laboratory closures, tissues were preserved in RNA later (Qiagen) before storage at -80°C. In total, we obtained two to four replicates per sex and tissue (S1 Table).

## RNA extraction and sequencing

TRIzol solution (1 mL) and a small amount of ceramic beads (Sigmund Lindner) were added to each tube containing tissue. Samples were homogenized using a tissue homogenizer (Precellys Evolution; Bertin Technologies). Chloroform (200 μL) was added to each sample and samples were then vortexed for 15 seconds and centrifuged for 25 minutes at 12,000 revolutions per minute (rpm) at 4°C. The upper phase containing the RNA was then transferred to a new 1.5 mL tube with the addition of isopropanol (650 μL) and Glycogen blue (GlycoBlue Coprecipitant; 1 μL). The samples were vortexed and placed at -20°C overnight. Samples were then centrifuged for 30 minutes at 12,000 rpm at 4°C. The liquid supernatants were removed, and the RNA pellet underwent two washes using 80% and 70% ethanol. Each wash was followed by a 5-minute centrifugation step at 12,000 rpm. Finally, the RNA pellet was resuspended in nuclease-free water and quantified using a fluorescent RNA-binding dye (Quanti-Fluor RNA System) and nanodrop (DS-11 FX).

Library preparation using NEBNext (New England BioLabs) and sequencing on an Illumina NovaSeq 6000 platform with 100 bp paired-end sequencing (~45 million read pairs per sample on average) was outsourced to a sequencing facility (Fasteris, Geneva).

## Data analyses

**Read counting.** Aligned RNA-Seq reads, sorted by genomic position, were processed to count reads mapped to exonic regions using HTseq [48] (v.011.2, options: --order=pos --type=exon --idattr=gene_id --stranded=reverse).

**Dosage compensation analysis.** We categorized the data based on tissue and developmental stage. Subsequently, we excluded genes with low expression, specifically those that were not expressed in a minimum of two libraries and with expression levels lower than 0.5 CPM (counts per million) across samples within the specified tissue and stage. Upon filtering, the number of genes kept for analysis ranged between 12143 and 14424 in different

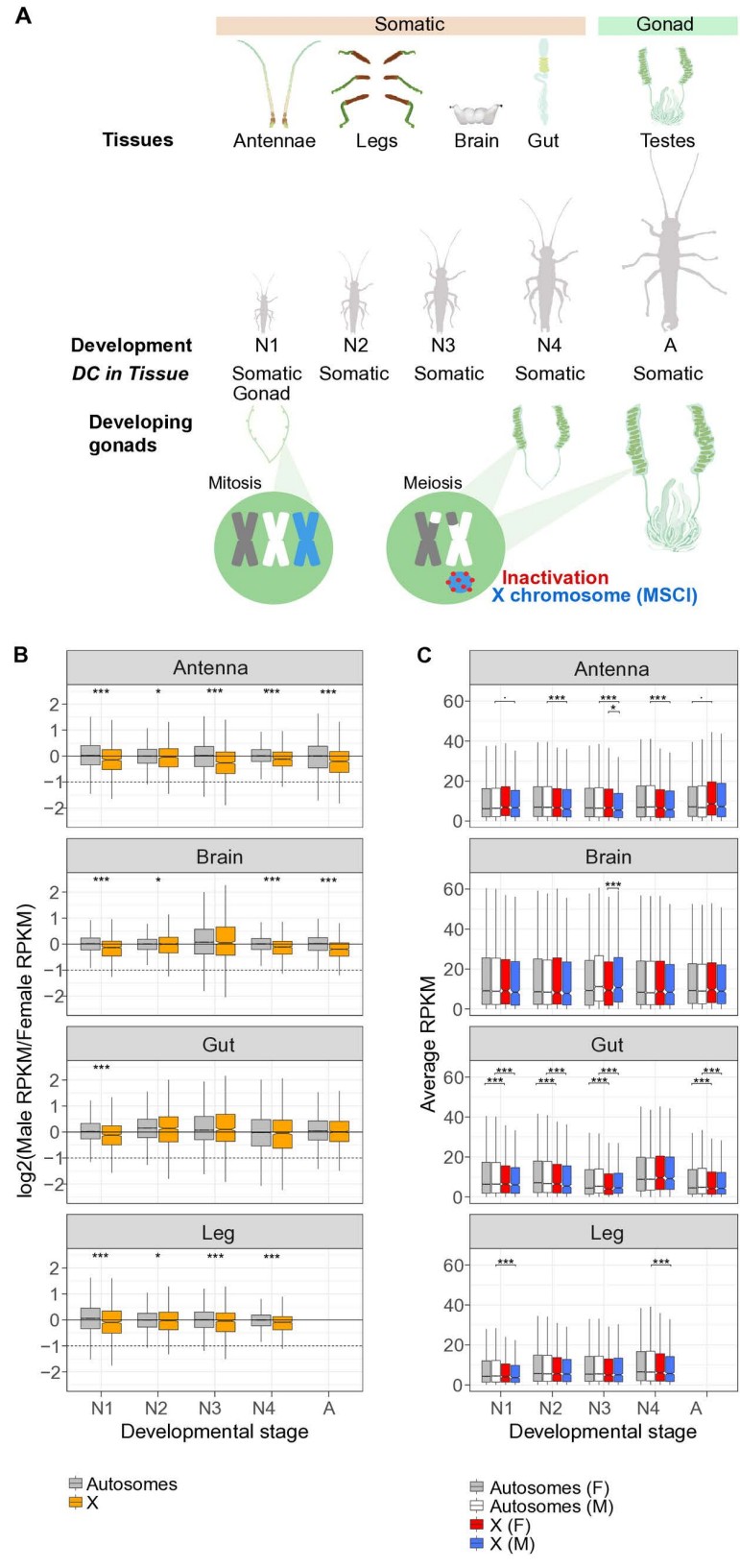

**Fig 1. Complete dosage compensation in male somatic tissues during development.** A) Schematic overview of male developmental stages in *Timema* stick insects (hemimetabolous development), different analyzed tissues, and

main findings. B) Log2 of male to female expression ratio for the X (orange) and autosomes (grey) in somatic tissues along development. Dashed lines represent a two-fold reduction in expression in males (as expected if there was no dosage compensation). See S2 Fig for individual rather than pooled autosomes C) Average expression levels (RPKM values) of genes located on the autosomes and X in females (grey and red) and in males (white and blue) in somatic tissues during development. Boxplots depict the median, the lower and upper quartiles, while the whiskers represent the minimum and maximum values, within 1.5x the interquartile range. See S3 Fig for individual rather than pooled autosomes. Significance symbols denote: ***= p < 0.01, *= p < 0.05, = p=0.05.

tissues and stages. Samples were then normalized for their library sizes using the function "calcNormFactors" in EdgeR [49].

For the computation of RPKM values, we first obtained gene sizes using the Genomic-Features package [50] and then calculated the average RPKM values for each sex using the "rowMeans" function in EdgeR [49]. Lastly, we determined the log2 RPKM ratio between the male and female average expression levels. Statistical analyses (Wilcoxon tests) and graphical representations of the data were performed in R (4.3.1).

**Tissue specificity (Tau).** To test if the X chromosome has more tissue-specific expression than the autosomes, we calculated Tau, an index of gene expression tissue specificity ranging from zero (expressed equally in all studied tissues) to one (gene expressed in only one tissue) [51]. For these analyses we used the subset of four tissues (antenna, brain, gut and reproductive tract) and four developmental stages (1st, 3rd, 4th nymphal and the adult stage) for which we had data for both sexes (see S1 Table). Using median expression RPKM values, we calculated Tau for the 12 longest scaffolds (corresponding to the 12 chromosomes of *T. poppense*). We visualized the results with ggplot2 (v. 3.3.2) [52].

**Immunofluorescence staining of male meiotic cells.** We assessed X transcriptional activity and heterochromatization in male meiotic cells via immunofluorescence staining. In order to determine the presence of meiotic cells, we used SMC3_488 (#AB201542, AbCam; hereafter SMC3) as a marker of cohesin axes (this allows for the assessment of synapsis progression, and hence, cell cycle). We performed double immunolabeling of SMC3 with RNA polymerase II phosphorylated at serine 2 (p-RNApol-II-S2P; ab193468, AbCam), an indicator of transcriptional activity, and with H3K9me3 (tri-methylation of histone H3 at lysine-9; #AB8898, AbCam), a histone modification generally associated with heterochromatic silencing, and which marks the silenced X chromosome in male mouse spermatogenesis [53,54]. Gonads from 1st and 4th nymphal stage males were dissected in 1X PBS and fixed in paraformaldehyde (2%) and Triton X-100 (0.1%) solution for 15 minutes. Gonads were then squashed on slides coated with poly-L-lysine, and flash frozen in liquid nitrogen. Slides were incubated in PBS 1x for 20 minutes at room temperature, then blocked with a BSA 3% solution (prepared by dissolving 3g of BSA in 100mL of PBS 1X) for 20 minutes. Slides were then incubated overnight at 4°C in a humid chamber with the primary (p-RNA-Pol II or H3K9me3), diluted in BSA 3% (1:100), and washed 3x for 5 minutes in PBS 1X. The secondary antibody, anti-rabbit Alexa 594 (711-585-152, Jackson), diluted in BSA 3% (1:100), was applied, and samples were incubated for 45 minutes at room temperature. Further washes were conducted as described above, followed by an extended 10-minute wash in PBS 1X, and by blocking using a 5% Normal Rabbit Serum (NRS) solution (X090210-8, Agilent Technology) for 30 minutes at room temperature. Another 5-minute wash in PBS 1x was performed to remove excess blocking solution. Samples were then incubated with the labeled antibody SMC3_488 (#AB201542, AbCam) at a 1:100 dilution for 1 hour at room temperature, followed by a final series of washes and staining with DAPI (D9542, Sigma-Aldrich) for 3 minutes at room temperature. Finally, after a 5-minute wash in 1x PBS, two drops of Vectashield were added to the slide, which was then covered with a coverslip, and

sealed with nail varnish for imaging. Image acquisition was performed at the Cellular Imaging Facility (CIF, University of Lausanne) using a Zeiss LSM 880 Airyscan equipped with a 60x/oil immersion objective. Image processing (cropping and pseudocoloring), was carried out using Fiji [55].

**ChIP-sequencing to reveal heterochromatin marks.** To profile silencing histone modifications on the X chromosome in male gonads and corroborate that the strongly heterochromatized chromosome revealed by immunofluorescence staining (see results) was in fact the X chromosome, we conducted ChIP-sequencing using H3K9me3 (based on the same antibody as used for immunofluorescence staining). We dissected male gonads (~ 40 individuals) and immediately froze them in liquid nitrogen. For chromatin preparations, the pooled frozen gonads were homogenized by cryogenic grinding (CryoMill; Retsch GmbH) using a specific regimen (2x 60 s, 25 Hz, resting 30 s, 5 Hz), transferred to a 15-ml Falcon tube, and subsequently subjected to rotation at room temperature for 12 minutes in a cross-linking solution composed of 50 mM Hepes (pH 7.9), 1 mM EDTA (pH 8), 0.5 mM EGTA (pH 8), 100 mM NaCl, and 1% formaldehyde. The cross-linking reaction was stopped by pelleting nuclei for 2 min at 2000g, followed by rotation for 10 minutes in a stop solution containing PBS, 125 mM glycine, and 0.01% Triton X-100. Nuclei were then subjected to washing steps in solution A [10 mM Hepes (pH 7.9), 10 mM EDTA (pH 8), 0.5 mM EGTA (pH 8), and 0.25% Triton X-100] and solution B [10 mM Hepes (pH 7.9), 1 mM EDTA (pH 8), 0.5 mM EGTA (pH 8), 0.01% Triton X-100, and 200 mM NaCl]. This was followed by a sonication step in 100 μl of radioimmunoprecipitation assay (RIPA) buffer [10 mM tris-HCl (pH 8), 140 mM NaCl, 1 mM EDTA (pH 8), 1% Triton X-100, 0.1% SDS, 0.1% sodium deoxycholate, and 1× complete protease inhibitor cocktail] in AFA microtubes in a Covaris S220 sonicator for 5 min with a peak incident power of 140 W, a duty cycle of 5%, and 200 cycles per burst. Sonicated chromatin was centrifuged to pellet insoluble material and snap-frozen.

ChIP was carried out using 5 μl of H3K9me3 (#AB8898, AbCam) incubated overnight at 4°C with half of the prepared chromatin sample (10 ul of the same chromatin preparation was used as input control, see below). Protein A Dynabeads (50 μl; Thermo Fisher Scientific, 100-01D and 100-03D) were added for 3 hours at 4°C, and subsequently washed for 10 min each once with RIPA, four times with RIPA with 500 mM NaCl, once in LiCl buffer [10 mM tris-HCl (pH 8), 250 mM LiCl, 1 mM EDTA, 0.5% IGEPAL CA-630, and 0.5% sodium deoxycholate], and twice in TE buffer [10 mM tris-HCl (pH 8) and 1 mM EDTA]. DNAs of the ChIP sample and the input control were then purified by ribonuclease digestion, pro-teinase K digestion, reversal of cross-links at 65°C for 6 hours, and elution from a QIAGEN MinElute PCR purification column. The purified DNA (ChIP and input) was then processed at the Lausanne Genomic Technologies Facility for library preparation using the NEBNext Ultra II DNA Library Prep Kit for Illumina and 150-bp paired-end sequencing on an Illumina HiSeq 4000.

ChIP and input reads were trimmed using trimmomatic (v0.39). The reads were then mapped to the reference genome using the BWA-MEM algorithm v0.7.17 (-c 1000000000), with ~57 million reads mapped and with a total mapping efficiency of 99.57%. Chimeric reads were removed using SA:Z tags, and PCR duplicates were eliminated with Picard tools (v2.26.2). Mean coverage was computed for ChIP and input reads within non-overlapping 30 kb windows across all scaffolds using BEDTools (v2.30.0) and normalized by the number of mapped reads in each library. For each genomic window, we computed the log2 ratio of H3K9me3 ChIP to input coverage. We visualized the data using the ggplot2 package in R (v3.42.4). Smoothing lines were added using a Generalized Additive Model (GAM) to high-light signal strengths across the genome, with smoothing applied individually to each scaffold to visualize differences between chromosomes.

## Results and discussion

### Complete dosage compensation across male somatic tissues and developmental stages

*T. poppense* stick insects have 12 chromosomes, comprising one X chromosome (third in size) and 11 autosomes [25]. Our genome assembly fully matches the structure expected from karyotypes, with 1.26 Gbp (94%) of the 1.34 Gbp assembly comprised in the 12 longest scaffolds (hereafter chromosomes) and scaffold 3 corresponding to the X chromosome (132.38 Mbp and harboring ~6% of total genes; S1 Fig). In all four somatic tissues examined (guts, brains, antennae, legs), males feature complete dosage compensation throughout nymphal development as well as at the adult stage. This is revealed by similar ratios of male to female expression for autosomal and X-linked genes (Fig 1B, S2 Table). Additionally, gene expression levels are relatively constant along the single male X (S2 Fig). This indicates that dosage compensation is achieved through a global mechanism that affects the entire chromosome.

While dosage compensation is complete between males and females, sometimes referred as dosage balance [56] (Fig 1B), X-linked genes generally have significantly lower expression than autosomal genes in male somatic tissues (Fig 1C; S3 Table). A similar pattern is observed for females (Fig 1C), with a significant reduction of X-linked expression in female gut and gonad tissue; (S3 Table). It can be argued that complete dosage compensation should refer to situations where the single X is transcribed at a level comparable to the ancestral levels for two X copies, prior to X chromosome formation (i.e., when the X chromosome would have been an autosome; [56]). However, the X chromosome in stick insects has been conserved for at least 120 Mya (within the Phasmatodea order [13,57]) and shares most of its content with the X chromosome in other insect orders that diverged over 450 million years ago [15]. This very deep conservation makes the inference of the ancestral transcription levels challenging.

Independently of the most appropriate terminology, the reduced expression of X-linked as compared to autosomal genes in females might indicate that the mechanism of dosage compensation in stick insects involves downregulation of the X chromosomes in females, similar to *C. elegans*. Alternatively, there may also be a limit to how much the transcriptional activity of the single X in males can be increased. This would favor the movement of highly expressed genes from the X to autosomes [58,59]. A limit to the maximal transcriptional activity of the single X in males is notably believed to explain X chromosome gene content and reduced expression levels in therian mammals [58,59] and *Drosophila* fruit flies [60]. Support for a similar explanation in *Timema* stems from the increased tissue-specificity of X-linked genes in both male and female *Timema* (Wilcoxon test, $p_{adj\ (females)}$ = 3.5e-11, $p_{adj\ (males)}$ = 2.3e-08; Fig 2; see also S4 Fig and S5 Table). This mirrors findings in mammals where tissue-specific genes are generally less highly expressed than broadly expressed genes [59,61], although opposite patterns are reported in *Drosophila* [62,63]. However, it is important to note that dosage compensation is a feature of individual cells and that a greater expression breadth across tissues does not necessarily equate with a higher expression level within individual cells, as would be the case if there was high heterogeneity in expression among cells within a tissue.

Additionally, X chromosome degradation could also contribute to lower expression of X-linked as compared to autosomal genes, similar to patterns reported for genes on the Y chromosomes in some species [64]. The X chromosome has a reduced effective population size as compared to autosomes, as a consequence of reduced copy numbers and the lack of X recombination in males, which leads to the accumulation of deleterious mutations in *Timema* [13].

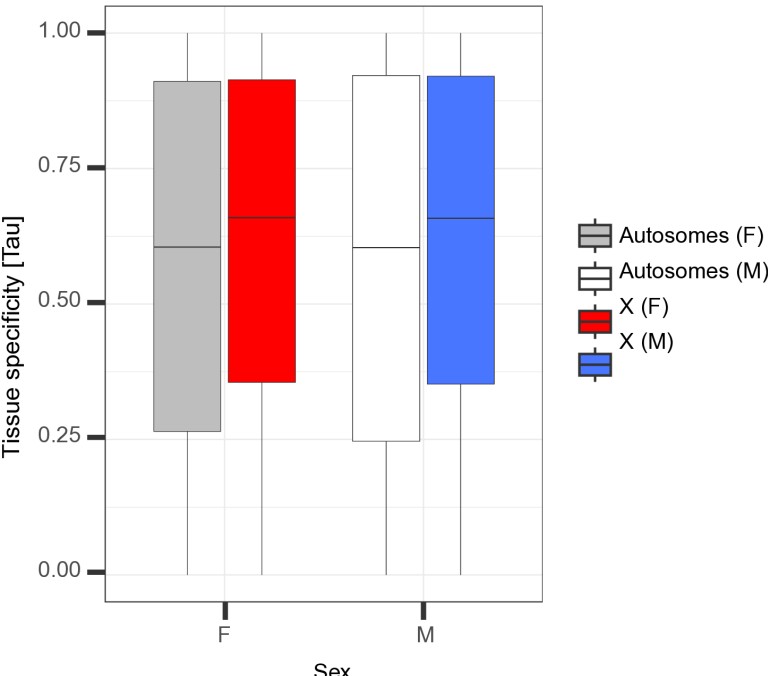

**Fig 2. X-linked genes are more tissue-specific in their expression than autosomal genes.** Tissue specificity of expression for genes on the X chromosome in males (blue box) and females (red box), compared to autosomal genes (white and gray boxes), calculated across four tissues. Boxes depict the median, the lower and upper quartiles, while the whiskers represent the minimum and maximum values, within 1.5x the interquartile range. Note that this pattern is not solely driven by averaging values across 11 autosomes or gene expression in gonads (S4 Fig).

## Early presence and later absence of dosage compensation in the male reproductive tract

The patterns of X-linked expression in *Timema* male reproductive tracts differ strikingly from those observed in somatic tissues. During the first nymphal stage, there is complete dosage compensation in the reproductive tract, similar to somatic tissues (Fig 3A–3C, and S4 Fig). However, from the 4th nymphal stage, there is a strong reduction of X-linked expression in males, relatively constant along the length of the X (S6 Fig), and which persists throughout development and until the adult stage (Fig 3). We first hypothesized that dosage compensation might be active during early gonad development in the first nymphal stage because the tissue was not yet strongly sexually differentiated. However, this is not the case as the degree of gonadal sexual differentiation (as measured from sex biased gene expression) is relatively constant throughout development. Already 44% of the expressed genes in gonads show sex-biased expression at the first nymphal stage, as compared to 45% at the 4th nymphal stage or 53% in adults (S4 Table).

Next, we evaluated whether the strong reduction of X-linked expression from the 4th nymphal stage was caused by meiotic X-chromosome inactivation, as is well described in therian mammals [65]. Indeed, the extent of the reduction significantly exceeds the two-fold reduction expected in the absence of dosage compensation (Wilcoxon signed rank test with continuity correction; 4th nymphal stage: V = 77366, p-value <2.2e-16, adult stage: V = 150046, p-value <2.2e-16) indicating other mechanisms are affecting the expression of X linked genes in male gonads (Fig 3A and 3B). Immunostaining of gonad tissue from adult and 4th nymphal stage males corroborates the transcriptional inactivity of the X chromosome. The X

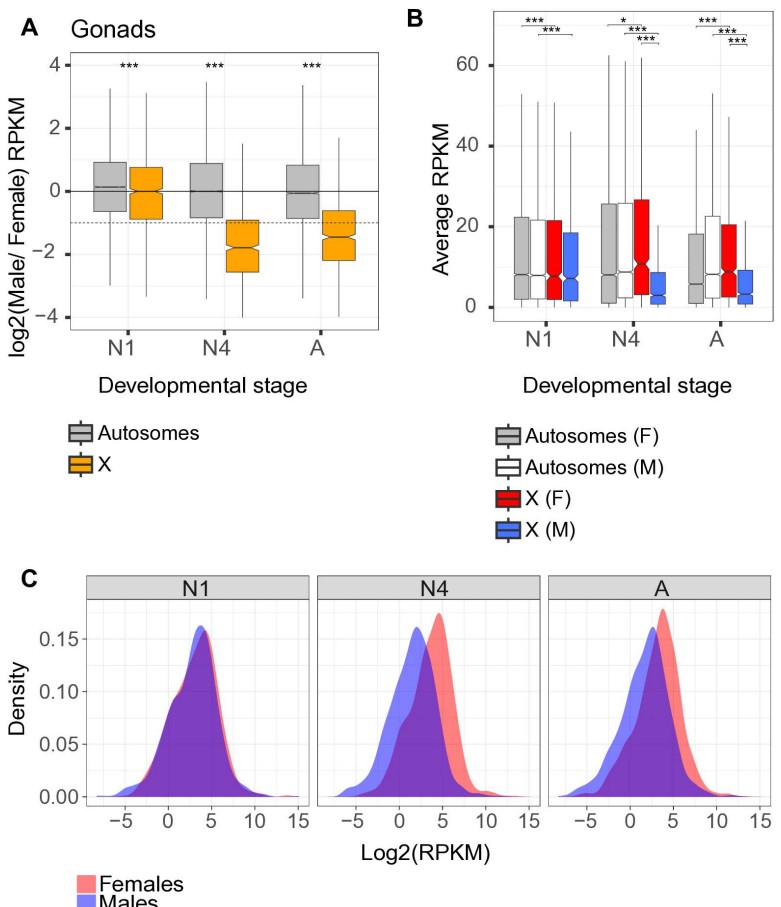

**Fig 3. Presence of dosage compensation only during the 1st nymphal stage in male reproductive tracts.** A) $\text{Log}_2$ of male to female expression ratio for the X chromosome (orange) and autosomes (gray) at three developmental stages (N1- 1st nymphal, N4- 4th nymphal and A- adult stage) in the reproductive tract. See S5 Fig for individual rather than pooled autosomes. B) Average expression levels (RPKM values) of genes located on the autosomes and X in females (gray and red boxes) and in males (white and blue) at three developmental stages. Boxplots depict the median, the lower and upper quartiles, while the whiskers represent the minimum and maximum values, within 1.5x the interquartile range. See S5 Fig for individual rather than pooled autosomes. C) Distribution of expression levels for X-linked genes in males (blue) and females (red) (overlapping ranges are indicated in purple). Significance symbols denote: ***= $p < 0.01$, *= $p < 0.05$.

chromosome in male meiotic cells is visible in DAPI staining as highly condensed ("hetero-pycnotic"; Fig 4), similar to other insects [66]. It also lacks a Pan Phospho RNApol-II signal, a marker for transcriptional activity, in otherwise transcriptionally active cells (Fig 4). We then also performed immunostaining of meiotic cells for silencing histone modifications (using an H3K9me3 antibody) and profiled the corresponding molecular marks in the genome using ChIP-seq based on male gonad tissue. As expected, the X has a strong H3K9me3 signal (Fig 5A and 5B) and the corresponding H3K9me3 marks are enriched all along the male X (Fig 5C), suggesting that the silencing of the X is achieved via histone modifications (Fig 5).

The patterns of MSCI in stick insects share many parallels with MSCI in mammals and birds [67] but differ strikingly from meiotic X chromosome regulation in the more closely related fly species *D. melanogaster*. Indeed, whether there is some form of targeted X chromo-some downregulation in male meiotic cells in *D. melanogaster* is a matter of ongoing debate, although it has become clear that MSCI as known for mammals and now stick insects does

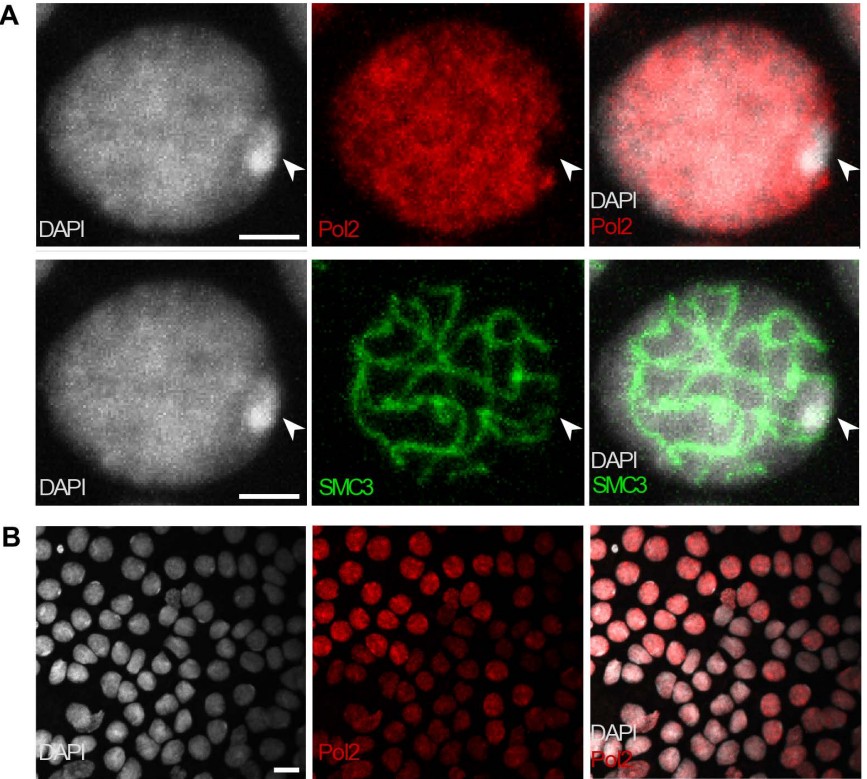

**Fig 4. No transcriptional activity of the X chromosome during male meiosis.** DAPI, Phospho S2 RNA Pol-II (Pol2-S2P), and SMC3 stainings were applied to germ cell squashes from 4th nymphal stage males, shown as an overlay of multiple focal planes. A) Shown is a cell at the pachytene stage of meiosis I for illustration. The arrow in each image indicates the location of the X chromosome (heteropycnotic body). The Phospho S2 RNA Pol-II staining highlights phosphorylated forms of RNA polymerase II, indicating regions of transcriptional activity. Scale bar indicates 5μm. B) Zoom out to illustrate the widespread transcriptional activity in meiotic cells and systematic lack of activity on the X chromosome. Scale bar indicates 20μm.

not occur in *D. melanogaster*. This is revealed by transcriptional profiling of *D. melanogaster* testis [68–70], similar RNA polymerase activity on the X and autosomes [71], and lack of enrichment of silencing histone modifications on the X in the male germline [71].

The difference between stick insects and *Drosophila* regarding X chromosome regulation in male germ cells is interesting because it can help elucidate the reasons for why MSCI evolved and was then secondarily lost or highly modified in a subset of species. Indeed, a possible explanation for MSCI is that it evolved to protect un-synapsed chromosomes (including the X in males) from damaging effects such as ectopic exchanges, non-homologous recombination, and unrepaired double-strand breaks [72]. The tight condensation of the X under MSCI would then shelter the unsynapsed X from such effects. A key difference between male meiosis in *Timema* stick insects and *D. melanogaster* is that meiosis is achiasmatic in *Drosophila* males [73] but chiasmatic in *Timema* [13]. The evolution of a completely achiasmatic meiosis in *Drosophila* males means that all chromosomes, not only hemizygous sex chromosomes, would be exposed to damaging effects in the absence of synapsis. Such a situation should generate strong selection for sheltering mechanisms not linked to hemizygosity [72] and could thus explain the loss of classical MSCI.

An association between achiasmy and lack of classical MSCI via strong X condensation is supported beyond the *Timema* vs *Drosophila* dichotomy by a survey of older cytogenetic

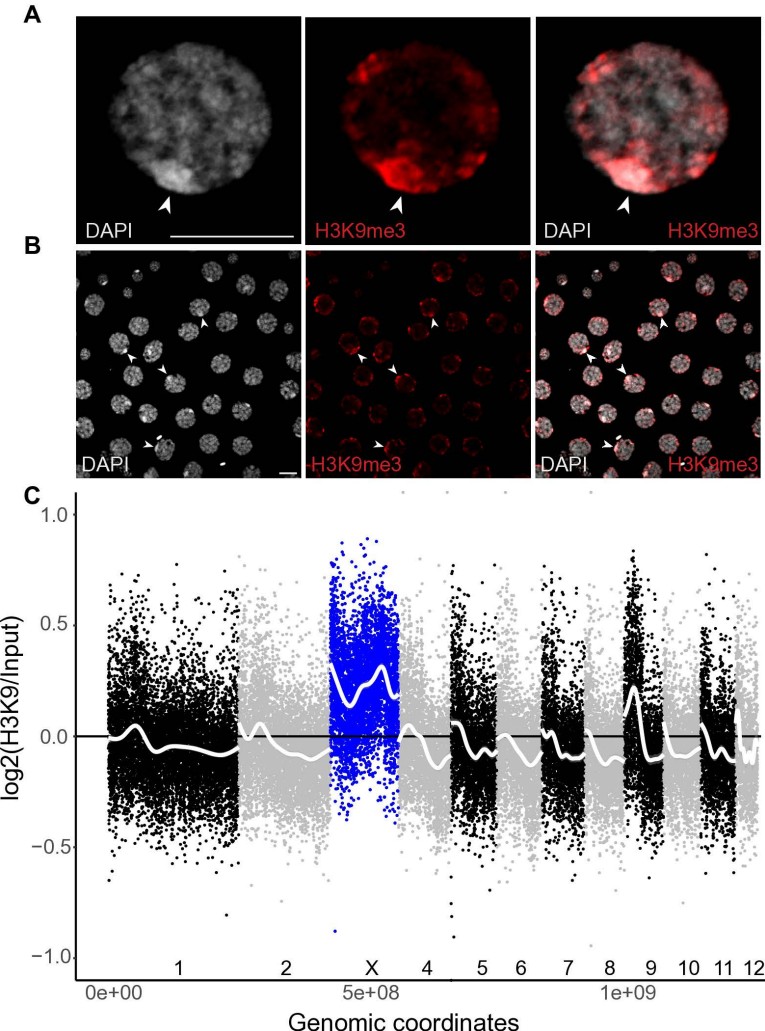

**Fig 5. Enrichment of the H3K9me marks on the X chromosome in adult male testes.** A, B) DAPI (blue) and H3K9me3 (magenta) applied to cells from male adult gonad squashes. Images from a single focal plane, of one individual A) single germ cell at pachytene stage, with X indicating the location of the X chromosome (scale bar 10μm) B) Many germ cells, with arrows pointing to the X chromosome C) Log2 ratio of H3K9me to input ChIP-Seq data along the genome, 30K window size. The X chromosome is depicted in blue, while autosomes are depicted in black or gray, and chromosome number (1-12) is further indicated above the Y axis. Smoothing lines (method "gam") for each scaffold are in white.

data in insects. Chiasmatic insects, including most species in the orders Orthoptera, Blattodea, Mantodea and the suborder Heteroptera, typically show sex chromosome morphologies that likely reflect heterochromatization (i.e., MSCI), while many (though not all) achiasmatic insects do not. This pattern is consistent with the idea that achiasmatic insects evolved from ancestral, chiasmatic forms with MSCI; some achiasmatic groups have then subsequently lost MSCI [72].

Independently of the causes underlying the evolution and loss of MSCI, the contrasting X-expression patterns in male reproductive tracts of hatchlings versus 4[th] nymphal stage likely reflect the start of meiosis I. Thus, in hatchlings where dosage compensation is complete (Fig 3A), cell divisions would be largely mitotic (spermatogonia stage). MSCI would start in cells entering meiosis I, resulting in the X-linked expression pattern we observe in reproductive

tracts at the 4th nymphal stage ([Fig 3A]). In accordance with these interpretations, observation of cells from 4th nymphal stage testes reveal the presence of many cells at meiosis I, whereas we could not find a single cell in meiosis I in squashes from testes of male hatchlings ([S7 Fig]). Finally, it appears that X-linked expression is somewhat elevated in reproductive tracts of adult males as compared to those of 4th nymphal stage males ([Fig 3A]). This is most likely due to the larger portion of somatic tissues in adult male reproductive tracts, most notably accessory glands, which are very small in all nymphal stages but well developed in adults (Jelisaveta Djordjevic, personal observation, see [Fig 1A]). Such a pattern is known to also occur in *Caenorhabditis elegans*, where the X is completely silenced in germ cells and changes in X-linked expression during development are caused by changes in the ratio of somatic to germ cells [56,74,75]. More generally, the observation of reduced X-expression in male gonads without investigation of associated mechanism is typically interpreted as lack of dosage compensation in this tissue [11–13,56]. However, as we observe in *Timema*, dosage compensation in males may well be ubiquitous across tissues and developmental stages in many species, and reduced X-expression patterns would solely be driven by MSCI.

## Conclusion

Dosage compensation by upregulation of the single X in males consistently occurs in somatic tissues across development in *Timema poppense*. By contrast, X-expression in male reproductive tracts is dynamic. It starts with complete dosage compensation in the first nymphal stage, when spermatogonia have not yet entered meiosis I, and is followed by X inactivation in later nymphal stages when most gonadal cells are undergoing meiotic divisions. We suggest that the reduced X-linked expression in testes is solely driven by X chromosome inactivation during meiosis, and does not stem from a lack of dosage compensation in *Timema*. This is likely generally the case in insects with reduced X-linked expression in testes.

## Supporting information

**S1 Table. Number of RNA-seq samples per sex, for every developmental stage and tissue.** (XLSX)

**S2 Table. Results of Wilcoxon tests where the ratio of male to female average RPKM values were compared between autosomes and the X chromosome within every developmental stage and tissue.** The column "stage" indicates the developmental stage; N1-N4 (1st to 4th nymphal stage), A- adult, the column "tissue" the different tissues, the column; "p.adjusted"-fdr correction of p-values for multiple testing using the Benjamini & Hochberg method (Benjamini and Hochberg 1995). (XLSX)

**S3 Table. Results of Wilcoxon tests where average RPKM values were compared for autosomal and X-linked genes for each sex and within each developmental stage and tissue (indicated in the "Tissue" column).** The "Stage" column denotes the developmental stage, with N1-N4 representing the 1st to 4th nymphal stages and A indicating the adult stage. **Columns: Group1 and Group2:** Specify the groups being compared, with colored cells highlighting the comparisons of interest. **"p.adjusted":** Represents the false discovery rate (FDR) corrected p-values, utilizing the Benjamini & Hochberg method for multiple testing. (XLSX)

**S4 Table. Sex-biased gene counts during reproductive tract development at N1 (1st nymphal), N4 (4th nymphal), and A (adult) stages.** It includes MB (male-biased), FB

(female-biased), SB (total sex-biased genes), and T (total expressed genes) categories, with percentages in brackets. Differential gene expression between the sexes was assessed using a generalized linear model with a quasi-likelihood F-test (Chen, et al. 2016) in edgeR v.3.42.4 (Robinson, et al. 2009; McCarthy, et al. 2012) in each of the three developmental stages as described in (Djordjevic, et al. 2022).
(XLSX)

**S5 Table. Wilcoxon tests comparing tissue specificity [Tau], calculated across three somatic tissues, between scaffolds within each sex ("variable"; Females, Males). Columns: Group1 and Group2:** Specify the scaffolds being compared, with colored cells highlighting the comparisons of interest, **"p.adjusted":** Represents the false discovery rate (FDR) corrected p-values, utilizing the Benjamini & Hochberg method for multiple testing.
(XLSX)

**S1 Fig. *T. poppense* X chromosome identification.** The plot shows the $\log_2$ ratio of male to female coverage of 100 kb sliding windows across the genome. Alternated colours designate different chromosomes, with chromosome 3 showing a much lower overall male to female coverage ratio.
(TIF)

**S2 Fig. Mean expression levels per gene along the X chromosome (in base pairs) in brain somatic tissue at 1st (N1), 4th (N4) and adult (A) stages in females (left) and males (right).** The line in each panel represents a loess (Locally Estimated Scatterplot Smoothing) smoothed curve, a non- parametric regression method that fits localized linear regressions to subsets of the data, highlighting overall expression patterns along chromosome.
(TIF)

**S3 Fig. Log2 of male to female expression ratio across all 12 *T. poppense* chromosomes at five developmental stages: N1, N2, N3, N4, and A (representing the 1st to 4th nymphal stages and the adult stage, respectively).** Chromosome 3, represented in orange, corresponds to the X chromosome, while autosomes are depicted in gray. The panels, arranged from top to bottom, showcase the Log2 ratio in different somatic tissues: antenna, brain, guts, and legs. Boxplots depict the median, the lower and upper quartiles, while the whiskers represent the minimum and maximum values, within 1.5x the interquartile range.
(TIF)

**S4 Fig. A) Average RPKM expression levels across chromosomes in females (grey) and in males (white) along development, chromosome three corresponds to the X and is depicted in red (females) and blue (males).** The panels, from top to bottom, showcase the Average RPKM in different somatic tissues: antenna, brain, guts, and legs. Boxplots depict the median, the lower and upper quartiles, while the whiskers represent the minimum and maximum values, within 1.5x the interquartile range. B) Distribution of expression levels for X-linked genes in males (blue) and females (red) (overlapping ranges are indicated in purple) across developmental stages and somatic tissues.
(TIF)

**S5 Fig. A) Tissue specificity of X chromosomes in males (M) (blue box) and females (F) (red box) as compared to the autosomes (white and gray boxes), calculated across three somatic tissues.** Because genes on the X are often testes or ovaries specific, we here repeated the analysis presented in the main text based on four tissues (three somatic tissues and reproductive tracts) with the three somatic tissues only, and X tissue specificity remained higher than autosomes Wilcoxon test, $p_{adj\ (females)}$= 6.7e-13, $p_{adj\ (males)}$= 2.4e-16 B) Tissue specificity in

females (F) and males (M) across scaffolds (see S5 Table), based on three somatic tissues. Scaffold 3, represented in orange, corresponds to the X chromosome, while other scaffolds are depicted in gray. Boxplots depict the median, the lower and upper quartiles, while the whiskers represent the minimum and maximum values, within 1.5x the interquartile range. (TIF)

**S6 Fig. A) The Log2 ratio of RPKM levels of expression between males and females across scaffolds at three developmental stages (N1, N4, and Adult) in the reproductive tract.** Chromosome three, represented in orange, corresponds to the X chromosome, while other chromosomes are depicted in gray. B) Average RPKM expression levels at three developmental stages (N1, N4, and adult) in the reproductive tract separated for different chromosomes in females (grey) and in males (white) boxes, chromosome three corresponds to the X and is depicted in red (females) and blue (males). Boxplots depict the median, the lower and upper quartiles, while the whiskers represent the minimum and maximum values, within 1.5x the interquartile range. (TIF)

**S7 Fig. Mean expression RPKM levels per gene along the X chromosome at 1st (N1), 4th (N4) and Adult (A) stages in reproductive tracts of females (left) and males (right) panels.** The red line in each panel shows the loess smoothed curve. (TIF)

**S8 Fig. DAPI, SMC3 staining applied to testes squashes of 1st (N1) and 4th (N4) nymphal stage males.** SMC3 is a protein of the cohesin complex marking chromosome axes during meiosis 1. No cells in meiosis I were detected at N1 (scale bar 10μm), while numerous cells at this stage were detected at N4 (scale bar 20μm). (TIF)

## Acknowledgments

We thank Corinne Peter and the Lausanne GTF platform for help with ChIP-seq, Bart Zijlstra, Armand Yazdani, Susana Freitas, and Chloé Larose for help in the field, and current and previous members of the Schwander lab for discussions.

## Author contributions

**Conceptualization:** Jelisaveta Djordjevic, Tanja Schwander.

**Data curation:** Jelisaveta Djordjevic, Jean-Marc Aury, Darren J. Parker.

**Formal analysis:** Jelisaveta Djordjevic, Patrick Tran Van, Darren J. Parker.

**Funding acquisition:** Tanja Schwander.

**Investigation:** Jelisaveta Djordjevic, William Toubiana, Marjorie Labédan, Zoé Dumas, Corinne Cruaud, Karine Labadie.

**Methodology:** Jelisaveta Djordjevic, Jean-Marc Aury, Benjamin Istace, Benjamin Noel, Tanja Schwander.

**Project administration:** Tanja Schwander.

**Supervision:** Tanja Schwander.

**Validation:** Tanja Schwander.

**Visualization:** Jelisaveta Djordjevic.

**Writing – original draft:** Jelisaveta Djordjevic, Tanja Schwander.

**Writing – review & editing:** Jelisaveta Djordjevic, William Toubiana, Zoé Dumas, Darren J. Parker, Tanja Schwander.

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
