## [Decision Letter · Decision Letter 0]

10 Sep 2024

Dear Dr Djordjevic,

Thank you very much for submitting your Research Article entitled 'Dynamics of X chromosome hyper-expression and inactivation in male tissues during stick insect development' to PLOS Genetics.

The manuscript was fully evaluated at the editorial level and by independent peer reviewers. The reviewers appreciated the attention to an important problem, but raised some substantial concerns about the current manuscript. Based on the reviews, we will not be able to accept this version of the manuscript, but we would be willing to review a much-revised version. We cannot, of course, promise publication at that time.

If you decide to revise the manuscript for further consideration at PLOS Genetics, please aim to resubmit within the next 60 days, unless it will take extra time to address the concerns of the reviewers, in which case we would appreciate an expected resubmission date by email to plosgenetics@plos.org.

If present, accompanying reviewer attachments are included with this email; please notify the journal office if any appear to be missing. They will also be available for download from the link below. You can use this link to log into the system when you are ready to submit a revised version, having first consulted our Submission Checklist .

PLOS has incorporated Similarity Check , powered by iThenticate, into its journal-wide submission system in order to screen submitted content for originality before publication. Each PLOS journal undertakes screening on a proportion of submitted articles. You will be contacted if needed following the screening process.

To resubmit, log into your Editorial Manager account and select the option 'Revise Submission' in the 'Submissions Needing Revision' folder.

We are sorry that we cannot be more positive about your manuscript at this stage. Please do not hesitate to contact us if you have any concerns or questions.

Yours sincerely,

Colin Meiklejohn

Academic Editor

PLOS Genetics

Monica Colaiácovo

Section Editor

PLOS Genetics

Dear Dr. Djordjevic,

Thank you very much for submitting your manuscript "Dynamics of X chromosome hyper-expression and inactivation in male tissues during stick insect development” (PGENETICS-D-24-00786) for consideration at PLOS Genetics. Your manuscript has been evaluated by three reviewers, all of whom recognized the novel and important contributions of your study. However, all the reviewers also raised points that need to be addressed before your manuscript could be considered for publication. A revised manuscript should address all the reviewer comments; in particular, the GitHub repository for the code and the NCBI SRA accession for the sequence data needs to be made available.

Reviewer's Responses to Questions

**Comments to the Authors:**

Reviewer #1: The manuscript by Djordjevic, Schwander and colleagues provides a timely characterization of X chromosome-associated gene expression changes in the stick insect Timema poppensis. To do this, the authors create a new chromosome-level assembly, followed by an analysis of sex-specific gene expression across developmental stages and tissues with the finding that dosage compensation is complete across tissues and stages. The authors then investigate the male germline more specifically and discover the presence of meiotic sex chromosome inactivation. This latter part is conclusive based on 1) the fact that repression goes way beyond the 2-fold difference expected solely due to the lack of dosage compensation (authors asses it by RNA-seq) 2) ChIP-seq of H3K9me3 showing enrichment of this repressive mark in male germline and 3) immunostainings in germ cells showing DAPI dense and Pol2-depleted nuclear areas.

Although there is no cytological proof that these dense areas are the X, the combination with ChIP-seq make it a well-supported evidence for MSCI occurring in Timema. The latter part is especially interesting because the male Drosophila germline does not exhibit MSCI, raising questions of how widespread this mechanism is and how quickly it evolves.

A highlight of the manuscript is its comprehensive sampling of transcriptome data across stages and tissues. While the manuscript itself focuses on dosage compensation and MSCI, this dataset will be certainly of great interest for a broad audience (for example, it could be used to study the evolution of gene expression across stages in different insects, or between sexes or between tissues). Meaning the dataset on its own is excellent and deserves publication in a journal like PLOS Genetics! I would also like to compliment the authors for their efforts in having a concise story with a clear focus on dosage compensation and thus, having “digestible” main figure panels. Another strength is that the authors attempt to complement their genomics analyses with cell biological work, which is not trivial in non-models such as Timema - protocols and antibodies are often unavailable / not easily adaptable.

I don’t think the paper has any experimental weaknesses but there are a couple of instances where there is a lack of clarity in data representation and/or writing and/or lack of context.

Here are some suggestions and comments that I think will make the paper more helpful for non-specialists in the field (unfortunately there were no line numbers in the manuscript, I hope it is clear to what I am referring to).

- Text: P.3/4 Introduction – There is some debate on whether mammalian X inactivation is a form of “dosage compensation”. I find this in the current intro text confusing considering that the second part of the manuscript deals with meiotic sex chromosome inactivation in males (but XCI in mammals occurs in females, while mammalian males have MSCI – the silencing of X-genes in the male germline thus a priori cannot occur due to a lack of the canonical X inactivation process which happens in females). Furthermore, C.elegans is a hermaphrodite and we generally have no idea how widespread the two-fold dampening is across taxa, because it has not been found elsewhere (in Danaus plexipus there is some suggestion on dampening, but not via H3K4me). These facts per se are interesting for our field, but I don’t think they need to be covered in the intro for this paper (besides that dosage compensation mechanisms are anyhow not the topic of the current study). I would recommend to focus the dosage compensation part in the intro on diversity in insects, maybe perhaps briefly stating mechanisms are different and mentioning that a current “hot” theme in the field is to investigate whether this process occurs in all tissues/dev stages. This would give a bit more space also to the discussion of ancient (non-dipteran insects) vs. different muller element becoming X in dipterans (the part about “derived” sex chromosomes later on this page is not well explained). Later when authors discuss the mosquito work, it sounds a bit like dosage compensation is regulated via alternative splicing the way the sentence is phrased. Alternative splicing regulates the sex-specific expression of SOA/007 protein in only males, but whether SOA/007 itself is a splicing regulator has not been investigated to date. The section at the beginning of p.4 is also not really clear, since male mammals have MSCI: X inactivation and its components are present in females, obviously this cannot explain why the X is repressed in the mammalian male testis (in males there is no Xist). In other words – I would recommend rewriting the introduction and focusing first on DC in insects (dipterans and non-dipterans) and MSCI. The presence of MSCI in insects as proven here is in my opinion also the most novel finding – besides the value of the dataset on its own (as stated above).

- Text, p.14 - “However, the X chromosome in stick insects has been conserved for at least 120 Mya [13, 53] and shares most of its content with the X chromosome in other insect orders that diverged over 450 million years ago [15], making it impossible to infer ancestral transcription levels.”

o Muller F in Diptera has reverted back to an autosome, so probably not meaningful for comparison. But what about other insects such as springtails? Or other insects (termites?) (see also Toups et al. paper)? I don’t mean that the authors need to do this analysis, but I would find it helpful to add this consideration in the text instead of giving such an absolute statement.

- Text, p.13.-14. I found the intermingling of discussion and results not optimally solved in this area of the manuscript. Perhaps this could be streamlined a bit better (better solved in later Figures/paragraphs).

- Text.p.20 – “The presence of MSCI in stick insects as we document here and lack thereof in D. melanogaster supports the first hypothesis. A key difference between male meiosis in Timema stick insects and D. melanogaster is that meiosis is achiasmatic in …” Very nice finding – the authors could stress that part even more (perhaps even in abstract) since they are of very general interest for the meiosis and DNA repair field.

Figures:

- Fig. 1 – Since stick insect development, the nymph-stages of hemimetabolous insects as well as germline development (meiosis at which stages?) are likely not a concept familiar to each reader – could the paper (Fig. 1A) start with an overview / schematic of the stages/tissues studied here, and key developmental transitions in the development of the germline (no meiosis / meiosis)?

- Fig. 1 – The y-axis scale is not helpful – one cannot really see the median line well (which is what matters here). Can the y-axis be separately scaled for each of the plots? Perhaps the authors can also provide density plots as in 3C (for example for the brain / antenna in adults, where there seems to be an interesting exception)? (along the same lines text, p. 14 “While dosage compensation is complete between males and females (Figure 1A), X-linked genes generally have significantly lower expression than autosomal genes in male somatic tissues (Figure 1B; Supplemental Table S3).” This cannot be really seen in the Figure. In my opinion the whiskers could be even removed, keeping simply the box.

- Fig. 2 – Interesting that the X shows more tissue-specific genes. Any preference for certain tissues here? I would find a bit more extra presentation on those results besides Tau index worthwhile.

- Fig. 4 – Immunofluorescence (IF): Which phosphorylation state of Pol2 does the antibody detect? Pan Phospho Pol2? Ser2P? Ser5P? Can this be clarified and also amended in the main Figure label in the IF? Also in the text on p.17 “It also lacks a RNApol-II signal, a marker for transcriptional activity, in otherwise transcriptionally active cells (Figure 4).” – please clarify accordingly.

- Fig. 5 – IF: From the methods it seems you have a double labelling for also SMC3 – can this data also be provided?

- Fig. 4/5 – can the style be unified? Some panels have no scale bar, sometimes a single scale bar without label of um, sometimes with label of um. Font size different, different arrow styles. Once magenta as color, once red. DAPI in blue, once in white. The overview Figures (B panel in Fig. 5) is missing in Figure 4 – this should be provided (especially for p-Pol2) – how often do you see the “hole” (I think I see it for DAPI in Fig. S8 - btw, here the labels are outside the figure, not inside, and scale bar is yet another style). A bit more effort on the consistent data representation in the IF would pay off.

- Fig. 5 – ChIP-seq

o Legend: “Smoothing lines (method “gam”) for each scaffold are in white.” – Can you please explain what this means in methods?

o What does the increase of signal towards the chromosome end mean? Is that subtelomeric heterochromatin? Can you pls explain?

o Can the x-axis be relabeled to the chromosomes and not as a single line? Why is the X blue? It was orange before.

Supplementary Information:

- Supp Figure S1 – what is Tps? What does the x-axis represent? Aren’t those the scaffolds from 1-12? Why representing this as one single axis in bp (the chromosomes have very different sizes, aren’t they?)

- Supplemental Table S3 – I appreciate the intention for full completeness of all tests performed – could the authors think of a way to include the significant comparisons also in the main figure?

- Supplementary Figure S1/ Legend – Can you explain what this means “The line in each panel represents a loess smoothed curve.”

- Supp. Figure S2 – is it bp?

- Supplemental Figure S3. / The authors have produced a beautiful chromosome-level assembly. Why keeping the nomenclature of scaffolds? I would recommend using chr1-chr12, and saying chr3 (X chromosome).

- Supp. Figure S3/S4 – y-axis are not ideally chosen (I guess it’s what comes out of R/ggplot2 by default) – however one can hardly see the differences / boxplots. I would manually adjust for each tissue so that the whiskers fit optimally in (e.g. ±2 for the Antenna in Fig. S3). Fig. S4 it’s impossible to see the median line. I find it well done / visible in S6A as a general guidance.

- Not sure how helpful the Supplementary Tables S2,S3,S5 are – they go across many pages making it really difficult to get an overview. Can one make a Supp. Figure instead and represent the raw values perhaps on another repository (github?)?

Methods / General:

- H3K9me3 ChIP-seq – can you provide a bit of QC please? Maybe I missed it but where is the info on replicates, number of mapped reads (%), where the enrichment is found (peak calling?)? Any other insights, except for the X?

- Generally letter panels (A, B, C) come in very different font styles (size, bold/not bold) – can you homogenize the layout? Also legends could sometimes profit from bigger font size.

- Methods: “Immunolabelling of X chromosomes” should be rephrased to “Immunfluorescence staining” (because you are not labelling the X chromosome here). Same paragraph, can you also give more details on the microscopy (magnifications, objectives, how you processed the images (Fiji?), do we see single planes or Max-intensity projections? How were they contrasted). How many nuclei / individuals / pictures were taken?

- For the dosage compensation analysis (methods, p.9): I did not quite understand what exactly has been done in edgeR and how the averages between males and females were calculated. How were for example replicates taken into account? How many genes did you compare versus each other? Is this the information provided in Supplemental Table S4? I think it would be nice to have this provided in methods as well as it looks a bit hidden / not easy to grasp at the moment. Can the authors provide all EdgeR results as supplementary Excel files (like what are the genes differentially expressed between males and females in the different stages, what are those, etc.)?

- I went to the github repository but was unable to find the code (neither through the link nor by going via the first author’s name).

- The Raw Sequencing Data could not be found on NCBI SRA with the accessions provided.

I apologize for the lengthy peer-review, but I think most of these issues are small, but relevant details that can be quickly and easily fixed by the authors.

Reviewer #2: Djordjevic et al. investigate dosage compensation (DC) in the stick insect Timema poppense, a species with X0 sex determination. They assemble a new genome sequence with long (ONP) and Hi-C reads, using the coverage difference between females and males to identify the X chromosome. By comparing RNA-seq data from both sexes, as well as from multiple tissues and developmental stages, they show that there is complete dosage compensation of the X chromosome in the somatic tissues of males. In male reproductive tissues of the 4th nymphal stage or adults there is no evidence of DC and average X-linked gene expression is reduced more than two-fold, suggesting that the X is actively silenced. This is supported by immunolabeling of male gonad tissue, which shows an absence of X transcription (RNA Pol2) and an enrichment of silencing histone modifications (H3K9me3). These results are consistent with the occurrence of MSCI (meiotic sex chromosome inactivation) in Timema.

The authors have done a thorough job investigating DC in a stick insect. It is especially notable that they have examined expression in multiple tissues and developmental stages and that they have performed immunohistochemistry to test for epigenetic marks associated with chromatin silencing. Below are a few suggestions that I think would improve the manuscript.

1) Abstract: "Stick insects are characterized by XX/XO sex determination and an X chromosome which likely evolved prior to the diversification of insects over 450 Mya" - would be better as "Stick insects are characterized by XX/XO sex determination, with an X chromosome that likely evolved prior to the diversification of insects over 450 Mya"

2) Page 12: "unambiguously identify the third largest scaffold as the X chromosome" - it would be helpful to mention how large this scaffold is and what proportion of genes in the genome are X-linked. It sounds like the X represents a larger proportion of the genome in Timema than in mammals or (maybe) Drosophila. The size of the X may play a role in the evolution of DC as the larger the X the more likely there are to be haploinsufficient genes that need to be compensated. Perhaps this contributes to differences between taxa?

3) The authors should indicate significant differences between X and A with asterisks (or "ns" if not significant) on figures 1, 2, and 3.

4) Some authors distinguish between dosage compensation (equalizing expression of X and autosomes in males) and dosage balance (equalizing expression of X between males and females). For example, see Gu et al., https://doi.org/10.1016/j.cub.2019.09.056. Since the authors test for both of these, it might be helpful to implement this terminology in their manuscript.

5) Page 14: To explain their observation that the X generally has lower expression than the autosomes, the authors state that "a limit to the maximal transcriptional activity of a haploid chromosome is notably believed to explain X chromosome gene content and reduced expression levels in therian mammals and Drosophila fruit flies". They then go on to support this hypothesis by noting that the X has higher tissue specificity than the autosomes in Timema and mammals, and that broadly expressed genes tend to have higher expression. I think there are a couple problems with this argument. First, in Drosophila the opposite pattern is observed: X-linked genes are less tissue specific than autosomal genes (Mikhaylova and Nurminsky, https://doi.org/10.1186/1741-7007-9-29; Meisel et al., https://doi.org/10.1101/gr.132100.111). Second, since dosage compensation occurs at the cellular level, it is not the total amount of expression in the organism that is relevant, but instead the amount per cell. Many tissue-specific genes have very high expression in cells of a particular tissue, but their overall expression level may be low when averaged over the entire organism.

6) Bottom of page 14: "Lastly, X chromosome degradation could also contribute to lower expression" - I'm not sure the paragraph needs to begin with "Lastly", as the authors don't really present a list of explanations in the preceding paragraph.

7) Bottom of page 17: I think the authors oversimplify the discussion regarding the presence/absence of MSCI in Drosophila. For example, studies that used reporter genes or transposed endogenous genes in which gene dose could be controlled showed clear evidence for suppression, but not complete inactivation, of X-linked gene expression in the male gremline (Hense et al., https://doi.org/10.1371/journal.pbio.0050273; Kemkemer et al., https://doi.org/10.1038/hdy.2013.86; Landeen et al., https://doi.org/10.1371/journal.pbio.1002499). scRNA-seq studies cannot control gene dose and, thus, cannot distinguish between the effects of lack of DC and those of MSCI in many cases. For example, ref. 51 cited in the manuscript seems to favor an X-inactivation model based on observations of the X and fourth chromosome, while the authors of ref. 52 explicitly state that they cannot make a conclusion about MSCI. The other cited references focus more on the mechanism to draw a conclusion, such as the absence of H3K9 methylation (ref. 64).

In the end, I think the authors conclusion mostly holds: in Timema, similar to mammals, there appears to be complete MSCI. The situation is different in Drosophila, where there does not appear to be complete MSCI, but rather partial suppression of X chromosomal gene expression in the male germline, which occurs through an unknown mechanism that does not involve H3K9 methylation. I suggest they alter their wording to reflect this.

Reviewer #3: Overall evaluation

This work addresses a model epigenetic process, dosage compensation, in non-model organism. The choice of an organism presumed to have an ancestral karyotype, and thus likely to retain an ancient mode of compensation, is particularly exciting. The authors have done a great deal of careful work, but there are logical lapses in presentation, and omission of critical details, that detract from impact and reader confidence.

Major points:

1) The authors assume that the mode of dosage compensation in T. poppense is increased expression from the male X chromosome, but this has not been demonstrated. Have the authors eliminated the possibility that both female X chromosomes decrease in expression? This is not a minor point. Even in the well-studied fly and mammalian systems the precise mechanism of achieving compensation is still being debated.

2) In A. gambiae, DC is regulated via sex-specific alternative splicing of the SOA gene, and activation of the male X occurs, although it does not involve H4K16ac. On page 3 it would be more appropriate to say that activation by SOA is poorly understood rather than unknown.

3) The authors do not confirm identity of the X in Fig. 4. As far as can be told from this manuscript the assumption is made that the condensed, DAPI-bright spot must be the X. I realize that confirmation may be beyond their expertise, but, if so, indicate that identity is presumed.

4) Indicate how many individuals/nuclei contributed to the histological identification of MSCI. How frequently was the SC and a DAPI-bright spot detected, and is Fig. 4 a representative image?

5) The authors claim that X-linked genes are expressed at lower levels in somatic tissues, but this is certainly not apparent in figure 1B. Table S3 is presented in support, but only contains p values. I am confident that the authors know their stuff but what is presented here does not enable the reader to reconstruct their argument. As a side note, low expression of X-linked genes in both sexes supports the idea that dosage compensation might occur through repression of both female X chromosomes – or by a combination of activation in males and repression in females. Can the authors eliminate these possibilities?

6) The onset of MSCI is nicely demonstrated by gene expression and quite convincing. But the legend to Fig. 3 states that compensation is absent in the late germ line, implying developmentally controlled dosage compensation. As the authors ultimately conclude that that reduced expression in male gonad in N4 is due to MSCI, a more nuanced figure legend would be appropriate.

7) Indicate number of replicates used for RNA sequencing.

Minor points

1) What stage are the germ cells in Fig. 5 A and B?

2) Edit manuscript for English usage and clarity throughout.

3) Discussion of chiasmatic vs. achiasmatic meiotic risk (bottom p. 20) hard to follow.

4) The blue fill in figures 1B, 2, 3B, S4, S5 and S6 is so dark that the median disappears.

**Have all data underlying the figures and results presented in the manuscript been provided?**

Reviewer #1: **No: ** Github codes and NCBI data could not be found.

Reviewer #2: Yes

Reviewer #3: Yes

PLOS authors have the option to publish the peer review history of their article (what does this mean? ). If published, this will include your full peer review and any attached files.

**Do you want your identity to be public for this peer review?** For information about this choice, including consent withdrawal, please see our Privacy Policy .

Reviewer #1: No

Reviewer #2: No

Reviewer #3: No

---

## [Decision Letter · Decision Letter 1]

14 Jan 2025

PGENETICS-D-24-00786R1

Dynamics of X chromosome hyper-expression and inactivation in male tissues during stick insect development

PLOS Genetics

Dear Dr. Djordjevic,

Thank you for submitting your manuscript to PLOS Genetics. After careful consideration, we feel that it has merit but does not fully meet PLOS Genetics's publication criteria as it currently stands. Therefore, we invite you to submit a revised version of the manuscript that addresses the points raised during the review process.

Please submit your revised manuscript within 30 days Feb 13 2025 11:59PM. If you will need more time than this to complete your revisions, please reply to this message or contact the journal office at plosgenetics@plos.org. Please include the following items when submitting your revised manuscript:

We look forward to receiving your revised manuscript.

Kind regards,

Colin Meiklejohn

Academic Editor

PLOS Genetics

Monica Colaiácovo

Section Editor

PLOS Genetics

Aimée Dudley

Editor-in-Chief

PLOS Genetics

Anne Goriely

Editor-in-Chief

PLOS Genetics

**Additional Editor Comments:**

While all of the reviewer comments are relatively minor, it is important that they be addressed in a revised manuscript, particularly the issues with ONT libraries in SRA, the availability of HiC and ChIP-Seq data, and actually revising the manuscript following the claims that were made in the authors' responses.

**Journal Requirements:**

Please ensure that the funders and grant numbers match between the Financial Disclosure field and the Funding Information tab in your submission form. Note that the funders must be provided in the same order in both places as well.

2) State what role the funders took in the study. If the funders had no role in your study, please state: "The funders had no role in study design, data collection and analysis, decision to publish, or preparation of the manuscript.".

**Reviewers' comments:**

Reviewer's Responses to Questions

**Comments to the Authors:**

Reviewer #1: I thank the authors for having provided a careful revision. This manuscript can now be accepted, but I recommend few amendments / details to be fixed:

The sex chromosome complement of Timema is sometimes referred to as X0 and sometimes as XO in the text. Please harmonize.

L70: Remove "Thus"

L104: Replace "immunolabeling" with "immunostaining" of "immunofluorescence staining"

L171: I recommend adding a title "Annotation generation" or something similar.

Methods talk of 7 ONT libraries, but the SRA data lists only 4. Also, the HiC raw reads data could not be found, neither is the ChIP-seq raw reads data available. Please upload/harmonize/clarify.

The authors seem to also have generated RNA-seq data from T. douglasi, but it´s not 100% clear to me from methods how these samples were collected (field? or lab population?). It would also be helpful in methods to quickly state how many MYA apart those species are.

I checked the antibody for Pol2, and it detects the Ser2P form.

Anti-RNA polymerase II CTD repeat YSPTSPS (phospho S2) antibody [EPR18855]

Can you correct the text accordingly, also label in the Figure (Pol2 should be replaced with Pol2-S2P)

Reviewer #2: The authors have addressed most of my concerns, although they were not always accurate or comprehensive with their revisions. There remain a few minor points that should be corrected before publication.

1) My previous comments on the abstract: "Stick insects are characterized by XX/XO sex determination and an X chromosome which likely evolved prior to the diversification of insects over 450 Mya" - would be better as "Stick insects are characterized by XX/XO sex determination, with an X chromosome that likely evolved prior to the diversification of insects over 450 Mya"

The authors wrote "changed as suggested" but it appears that they did not.

2) I appreciate that the authors indicated significance on their figures with asterisks, but they should also revise the figure legends to indicate what the asterisks mean. Is it *p<0.05, ** p<0.01, ***p<0.001?

3) The authors added the sentence "However, it is important to note that tissue specificity of expression (as well as dosage compensation) is a feature of individual cells, and that cell type composition varies across tissues. Biological effects can therefore be blurred when expression levels are averaged across all cells within a tissue." However, this is not quite correct and it also does not directly address the caveat to their interpretation. First, it does not make sense that "tissue specificity of expression ... is a feature of individual cells" as, by definition, tissue specificity of expression is a property of tissues. Second, the main point is that greater expression breadth across tissues does not necessarily equal greater expression within individual cells. So it would be better to write something like: "However, it is important to note that dosage compensation is a feature of individual cells and that a greater expression breadth across tissues does not necessarily equate to a higher expression level within individual cells, as would be the case if there was high heterogeneity in expression among cells within a tissue."

Reviewer #3: This is much improved but could still benefit from editing for language and clarity in several spots. Just a few that stood out to me are noted below.

line 75 change "in a diversity of . . " to " in diverse . . ."

Line 80-81 revise to clarify

364-374 revise to clarify.

Line 507 The idea underlying the final phrase " and most generally in insects." should be expanded as a stand alone sentence.

**Have all data underlying the figures and results presented in the manuscript been provided?**

Reviewer #1: Yes

Reviewer #2: Yes

Reviewer #3: Yes

PLOS authors have the option to publish the peer review history of their article (what does this mean? ). If published, this will include your full peer review and any attached files.

**Do you want your identity to be public for this peer review?** For information about this choice, including consent withdrawal, please see our Privacy Policy .

Reviewer #1: No

Reviewer #2: No

Reviewer #3: No

**Figure resubmission:**
---

## [Editor Report · Decision Letter 2]

5 Feb 2025

Dear Dr Djordjevic,

We are pleased to inform you that your manuscript entitled "Dynamics of X chromosome hyper-expression and inactivation in male tissues during stick insect development" has been editorially accepted for publication in PLOS Genetics. Congratulations!

Yours sincerely,

Colin Meiklejohn

Academic Editor

PLOS Genetics

Monica Colaiácovo

Section Editor

PLOS Genetics

Aimée Dudley

Editor-in-Chief

PLOS Genetics

Anne Goriely

Editor-in-Chief

PLOS Genetics

Comments from the reviewers (if applicable):

**Data Deposition**

http://datadryad.org/submit?journalID=pgenetics&manu=PGENETICS-D-24-00786R2

**Press Queries**

---

## [Editor Report · Acceptance letter]

PGENETICS-D-24-00786R2

Dynamics of X chromosome hyper-expression and inactivation in male tissues during stick insect development

Dear Dr Djordjevic,

We are pleased to inform you that your manuscript entitled "Dynamics of X chromosome hyper-expression and inactivation in male tissues during stick insect development" has been formally accepted for publication in PLOS Genetics! Your manuscript is now with our production department and you will be notified of the publication date in due course.

With kind regards,

Anita Estes

PLOS Genetics

On behalf of:
